# A Comparative Analysis of Strain and 2D Shear Wave Elastography in the Diagnosis of Autoimmune Thyroiditis in Pediatric Patients

**DOI:** 10.3390/biomedicines11071970

**Published:** 2023-07-12

**Authors:** Cristina Mihaela Roi, Andreea Borlea, Monica Simina Mihuta, Corina Paul, Dana Stoian

**Affiliations:** 1Department of Doctoral Studies, Victor Babes University of Medicine and Pharmacy, 300041 Timisoara, Romania; cristina.cepeha@umft.ro (C.M.R.); simina.mihuta@umft.ro (M.S.M.); 2Department of Internal Medicine II, Victor Babes University of Medicine and Pharmacy, 300041 Timisoara, Romania; stoian.dana@umft.ro; 3Department of Pediatrics, Victor Babes University of Medicine and Pharmacy, 300041 Timisoara, Romania; paul.corina@umft.ro

**Keywords:** strain elastography, shear-wave elastography, thyroiditis, children, thyroid stiffness, Hashimoto

## Abstract

This paper aims to assess the usefulness of shear-wave elastography (SWE) and strain elastography (SE) for identifying and monitoring thyroid gland changes in children diagnosed with chronic autoimmune thyroiditis (CAT). Our study included 77 children between the ages of six and eighteen. Of these, 45 were diagnosed with CAT, while 32 had no thyroid pathology. Following a clinical examination and laboratory tests, an ultrasound was carried out, and then a SE (using a Hitachi Preirus machine) and SWE (using an Aixplorer Mach 30, Supersonic imagine, France) were performed in the same session. The median thyroid elastic index (EI) in the CAT group was 13.8 (13.3–17) kPa compared to 10.1 (9.3–11.2) kPa in healthy children (*p* < 0.0001). We found a median strain ratio (SR) of 1.2 (1.2–1.3) for CAT compared to 0.7 (0.6–0.9) for healthy thyroid tissue (*p* < 0.0001). The optimal cut-off value for predicting the presence of CAT in children using SR was >1 (Se = 82.2%, Sp = 87.5%, PPV = 90.2%, and NPV = 77.8%, AUROC = 0.850), while using SWE, the optimal cut-off value for predicting the presence of CAT in children was >12 kPa (Se = 88.9%, Sp = 93.7%, PPV = 95.2%, and NPV = 85.5%, AUROC = 0.943). Both techniques are useful for measuring thyroid tissue elasticity, and their diagnostic accuracy and reliability are comparable.

## 1. Introduction

Elastography is a form of imaging that analyzes the elasticity or stiffness of tissues. It has become an essential diagnostic and monitoring tool for a variety of medical conditions. It relies on the concept that various tissues have distinct elastic properties. When a tissue is exposed to a mechanical force, its deformation depends on its elasticity. Using ultrasound, elastography analyzes this deformation and provides information about the elasticity of the tissue [1,2,3].

Strain elastography and shear wave elastography are the two primary forms of elastography. Strain elastography quantifies tissue deformation in response to an external force. It is frequently used in breast imaging to differentiate benign from malignant lesions [4]. Shear wave elastography measures the velocity of a shear wave generated by a minor tissue vibration. This form of elastography is useful for evaluating liver fibrosis [5], in addition to prostate [6] and thyroid nodules [7].

Elastography is utilized to distinguish between benign and malignant lesions. Typically, malignant tumors are more rigid than benign lesions, so elastography can guide biopsies. It is often used in liver imaging to evaluate fibrosis, an important complication of liver disease. Elastography can also be used to evaluate the rigidity of arteries, which can be a sign of cardiovascular disease [8].

Despite its numerous advantages, elastography has certain limitations. The technique is operator-dependent, which means that the image quality can vary depending on the operator’s experience. The presence of artifacts, such as calcifications or cysts, can also have an effect. In addition, the subjective nature of elastography image interpretation can lead to diagnostic variation [9].

Regarding thyroid examination, elastography has been demonstrated to be an effective method for evaluating common thyroid nodules in the general population. While the majority of thyroid nodules are benign, some can be malignant; therefore, an accurate diagnosis is essential for determining the most appropriate treatment. It can be used to evaluate the rigidity of thyroid nodules, which may help in discriminating between benign and malignant nodules. Malignant thyroid nodules are frequently more rigid than benign nodules, and elastography can provide additional diagnostic information. Studies have demonstrated that elastography is more accurate than conventional ultrasound for diagnosing thyroid nodules [10,11].

This technique can be used to monitor changes in the rigidity of thyroid nodules over time, in addition to its diagnostic utility. This is helpful when a nodule is initially determined to be benign but must be monitored for potential growth or rigidity change. It can provide additional information that may be useful in the diagnosis and monitoring of thyroid nodules, as well as guide their appropriate management [12,13]

Elastography can also be used to assess diffuse thyroid diseases like Hashimoto’s thyroiditis or Graves’ disease [14,15]. It evaluates the rigidity of the entire thyroid gland, which can help differentiate between healthy and diseased tissue. In cases of thyroiditis, an inflammation of the thyroid gland, elastography can distinguish between acute and chronic appearance of the disease and monitor disease progression [16].

It should be used alongside other diagnostic tools to provide a complete evaluation of diffuse thyroid diseases.

Chronic autoimmune thyroiditis, also known as Hashimoto’s thyroiditis, is a common thyroid gland autoimmune disorder, in which the immune system attacks the thyroid gland, resulting in inflammation and tissue damage [17]. Over time, this can lead to hypothyroidism [18]. Hashimoto’s thyroiditis can affect individuals of all ages, though it is more prevalent in middle-aged women. It is also more prevalent in families with a history of autoimmune diseases [19].

The signs and symptoms of hypothyroidism may be subtle and manifest progressively over time. These symptoms may include fatigue, weight gain, constipation, parched skin, and sensitivity to cold. In some cases, an enlarged thyroid gland may cause difficulty swallowing or discomfort. Blood tests and imaging investigations are typically combined to diagnose Hashimoto’s thyroiditis [20]. Imaging investigations, such as ultrasonography or elastography, can evaluate the size and texture of the thyroid gland. Typically, Hashimoto’s thyroiditis, if complicated with hypothyroidism, is treated with hormone replacement therapy [21]. To evaluate the efficacy of treatment and monitor for potential complications, such as the development of thyroid nodules or thyroid cancer, periodic monitoring of thyroid hormone levels and periodic imaging studies may be necessary.

Chronic autoimmune thyroiditis (CAT) is the most prevalent autoimmune disorder among children [22]. The precise prevalence of Hashimoto’s thyroiditis in children varies based on the examined population and diagnostic criteria. Hashimoto’s thyroiditis incidence increases with age and is more prevalent in girls than in boys [23]. The reported prevalence among minors is 4–10% but it may be higher in certain populations, such as those with Down syndrome or other genetic disorders [24,25]. The incidence varies from 1–2% in Italy to 4–5% in Denmark and up to 10% in Greece [22,26]. In addition to the typical symptoms of hypothyroidism, children may also experience delayed growth and development, as well as alterations in behavior or cognition. When repeated evaluations are necessary, non-invasive procedures are preferred. Elastography is one such reliable and effective method for assessing the thyroid.

The aim of this study is to evaluate and compare the diagnostic performance of strain elastography and 2D shear wave elastography in the detection of autoimmune thyroiditis in pediatric patients. This will be achieved by examining the sensitivity, specificity, and overall accuracy of each technique, as well as assessing their potential advantages and limitations in clinical practice.

## 2. Materials and Methods

### 2.1. Group Characteristics

The study was conducted from January 2022 to January 2023. It was performed according to the Helsinki Declaration’s Ethical Guidelines, and the local ethics committee approved it. Each parent of a child who participated in the study filled out an informed consent form. Our research included 77 children between the ages of 6 and 18 years old. Forty-five of them were diagnosed with CAT, while thirty-two did not have any thyroid pathology.

### 2.2. Inclusion and Exclusion Criteria

Children diagnosed with chronic autoimmune thyroiditis based on clinical examination, ultrasound appearance, and elevated levels of aTPO and aTG antibodies were included in our study. Of these, 19 of the 45 children diagnosed with CAT received hormone replacement therapy. Additionally in the control group were included children without any thyroid pathology, with normal laboratory tests and an ultrasound appearance suggestive of a normal thyroid. All subjects were examined at “Dr. D” Medical Center in Timisoara, Romania.

Excluded from the study were patients with nodular thyroid pathology, malignancies, or a history of thyroid surgery. Graves’ disease (GD) patients and those with acute or subacute thyroiditis were also excluded from the study. In addition, due to the difficulty of the evaluation, minors under the age of six were excluded from the study. Cases with ultrasound findings suggestive of CAT but normal antithyroid antibody titers were also excluded. The recruitment process of the subjects is illustrated in Figure 1.

### 2.3. Biochemical Assay

The following parameters were analyzed for each child: FT4—free-thyroxine (reference range 12–20 pmol/L; ECLIA method—immunochemistry with enzyme chemiluminescence immunoassay), TSH—thyroid stimulation hormone (reference range 0.5–4.80 µIU/mL; ECLIA method), ATG (reference range 35 IU/mL; ECLIA method), and ATPO (reference range 28 IU/mL; CMIA method—microparticle-based chemiluminescence immunochemistry). All biological analysis was conducted in an accredited laboratory.

### 2.4. Conventional Ultrasound and Elastography Examination

On a Hitachi Preirus machine with a 5–15 multifrequency linear transducer, conventional B-mode thyroid ultrasound and strain elastography (real-time elastography) were performed. The same practitioner evaluated all subjects by means of clinical examination and ultrasonography. To completely expose the neck, the optimal examination position was the supine position with the head tilted back. Patients were advised to avoid swallowing and speaking during the examination. Using grayscale ultrasound, the transverse and longitudinal diameters, thyroid volume, and echogenicity of the thyroid were measured (Figure 2) [27]. In the same visit, a conventional ultrasound was followed by real-time elastography. The probe was positioned perpendicular to the skin, and light, repetitive compression was applied without lateral movement. All images were captured in the longitudinal axis. A blue–green–red color map was displayed, with blue indicating no strain (high stiffness), green indicating intermediate stiffness, and red indicating soft tissue. To determine the strain ratio (SR), two regions of interest (ROI) were positioned consecutively using 15 MHz frequency. ROI A represented thyroid tissue, while ROI B represented the sternocleidomastoid muscle in front of the ipsilateral thyroid parenchyma. For each lobe, three consecutive measurements were taken, and the median value was used for the final evaluation. The SR was automatically calculated and displayed for each lobe. According to specialized literature, the use of SR has more specific results than the use of a color map alone [28,29].

Shear-wave elastography was performed during the same visit using the Aixplorer Mach 30 (Supersonic imagine, Aix-en-Provence, France) machine with an L 18-5 probe (linear, 5–18 MHz). The transducer was positioned on one side of the patient’s neck, and SWE mode was activated. The device displays a color map ranging from blue (indicating delicate tissue) to red (indicating rigid tissue). All images were captured along the longitudinal axis. Subjects were instructed to hold their breath for approximately five seconds. Therefore, the image was frozen, and tissue elasticity was measured with the Q-BOX and recorded in kilopascals. The region of interest (ROI) was positioned in the approximate center of the thyroid lobe. If hypoechoic regions were detected, the ROI was positioned in those areas. There were six measurements taken for each thyroid lobe of each subject. A quantitative value of elasticity was expressed in kilo-Pascals (kPa). The elastic index was determined using the EI mean in accordance with the guidelines [30]. All measurements were performed by the same practitioner with at least 3 years of experience in thyroid imaging.

### 2.5. Statistical Analysis

To summarize the characteristics of the study population, we performed descriptive and inferential statistical analyses. The D’Agostino–Pearson test was used to determine whether or not a dataset respects a normal distribution. This test for normality includes skewness and kurtosis. For reporting continuous variables with a non-normal distribution, measures of central tendency and measures of dispersion were used as well: the median value and the interquartile range, a robust measure of dispersion that covers the middle 50 percent of the data, between 25 and 75 percent. We applied the Mann–Whitney U test for numeric variables. The computation of the sample size was carried out with version 3.1.9.7 of the G*Power software, (Kiel, Germany). According to the calculation, a total of 45 participants would achieve a statistical power of 95%, a significance level of 5%, and an effect size value of 0.85. The diagnostic performance of the SWE EI in distinguishing CAT was evaluated using a ROC curve, and the thresholds for discriminating between CAT and normal thyroid were determined using Youden’s index. Pearson correlation was used to test the linear relationship between two normally distributed continuous variables, while Spearman rank correlation was used for non-normally distributed variables. A *p*-value less than or equal to 0.05 was recognized as statistically significant. Using Medcalc Statistical Package (version 12.5.0.0 64-bit), we analyzed the data.

## 3. Results

77 subjects were evaluated using both strain and shear-wave elastography measurements performed on the thyroid, with 45/74 (58.4%) diagnosed with CAT and 32/77 (41.5%) without thyroid pathology. In 77 out of 77 subjects, accurate measurements were obtained. In both categories, the percentage of women exceeded 75%, while the percentage of men was below 25%. The median TSH value for the CAT group was 3.4 (1.9–4) µIU/mL while it was 3.2 (2.4–3.9) µIU/mL for the control group (*p* = 0.752). The mean Ft4 value for healthy children was 13.6 (12.4–14.7) pmol/L, whereas it was 14.5 (12.8–15.1) pmol/L for the CAT group (*p* = 0.131). We found significant differences between the thyroid volumes of the children in the two groups (14.9 ± 6.5 mL for CAT vs. 9.3 ± 3 mL for the control group; *p* < 0.0001). The main characteristics of the included subjects are summarized in Table 1.

### 3.1. Strain Elastography Measurements

Figure 3 and Figure 4 show the SE image and measurement of the SR in normal thyroid tissue and in a patient with CAT, respectively. For strain evaluation, the median of three values was considered in the analysis, with no significant differences between the right and left thyroid lobes (U = 2864; *p* = 0.716). We obtained a median SR of 1.2 (1.2–1.3) for CAT vs. 0.7 (0.6–0.9) for healthy thyroid tissue (*p* < 0.0001) as shown in Figure 5.

### 3.2. SWE Measurements

A normal thyroid assessed via SWE is shown in Figure 6, while Figure 7 displays a thyroid affected by CAT. The median EI values for the left thyroid lobe were 13 kPa (10.7–14.7) and for the right thyroid lobe were 12.7 kPa (10.8–14.4) in the whole group: U = 2958; *p* = 0.981. There were no significant differences between these values. In the CAT group, the median thyroid EI was 13.8 (13.3–17) kPa vs. 10.1 (9.3–11.2) kPa in healthy children (*p* < 0.0001), as shown in Figure 8.

### 3.3. SWE vs. Strain

When both 2D-SWE and SE were carried out on both thyroid lobes of each child, it was discovered that there was no significant difference between the median values obtained in the left lobe and those obtained in the right lobe, respectively. Six measurements for each lobe using SWE and three measurements using SE were considered in our analysis (Table 2).

The optimal cut-off value for predicting the presence of CAT in children using SWE was >12 kPa (AUROC = 0.943, Se = 88.9%, Sp = 93.7%, PPV = 95.2%, and NPV = 85.5%) based on the median EI values and is illustrated in Figure 9. As shown in Figure 10, the optimal cut-off value for predicting the presence of CAT in children using SR was >1 (AUROC = 0.850, Se = 82.2%, Sp = 87.5%, PPV = 90.2%, and NPV = 77.8%). Comparing these two methods, we observed no significant differences between them (difference = −0.093; std error = 0.056; *p* = 0.101).

### 3.4. LT4 Replacement Therapy

A total of 19/45 (42.22%) minors in the CAT group were receiving LT4 replacement therapy. As shown in Table 3, there were no significant differences between the median EI and SR values of minors undergoing therapy and those without therapy.

### 3.5. Correlations

After statistical analysis, we found a very weak correlation between EI and BMI (r = 0.26), a weak positive correlation between EI and ATG values (r = 0.39), a moderate positive correlation between EI and TV (r = 0.43) and a strong positive correlation between EI and ATPO values (r = 0.66). We found no correlation between EI and age (r = 0.14), gender (r = 0.10), TSH (r = −0.09) or FT4 (r = 0.16).

A very weak correlation was found between SR and ATG values (r = 0.26). Weak positive correlations were also found between SR and BMI (r = 0.30) or TV (r = 0.30). Between SR and ATPO was found a moderate positive correlation (r = 0.46). No correlations were found between SR and TSH (r = −0.02), FT4 (r = 0.17), age (r = 0.14), or gender (r = −0.19). All studied correlations can be found in Table 4. No correlation was calculated between CAT and LT4 because it was considered not relevant.

## 4. Discussion

As one of the most prevalent endocrine disorders, CAT has been extensively studied over the years. Obesity and lymphocytic infiltration are two factors that can affect the ultrasound appearance of the thyroid. Thus, it is known that lymphocytic infiltration represents the histological substrate, with varying degrees of fibrosis contributing to the firm consistency of the thyroid. The various degrees of thyrocyte lesions produce inhomogeneity. Since inflammation causes fibrosis and inelasticity, it is worth investigating whether a greater degree of fibrosis indicates inflammation.

Both elastographic methods were studied both on the adult population and on the pediatric population, as well as on the thyroid and several other organs. Our study is the only one that we are aware of that compares these two approaches in the assessment of the thyroid in pediatric patients.

The amount of research undertaken on children is significantly lower compared to the number of studies completed on the population of adults; thus, this aspect makes our study even more significant.

In order to estimate the SR of normal thyroid parenchyma, Yurttutan et al. carried out research in which they recruited 54 healthy children to participate. The found mean value was 0.54 with a standard deviation of 0.38, similar to our results (0.7). There was found to be no association between SR and either age (r = 0.22; *p* = 0.15) or gender (r = 0.007; *p* = 0.96) [31].

To the best of our knowledge, there have been a total of just three studies in which children with CAT were examined using SE. A study that was carried out on 63 children who were diagnosed with HT and 47 children who did not have thyroid disease indicated that there were differences between the two groups (1.75 ± 1.46 vs. 0.26 ± 0.77; *p* 0.001), results that were comparable to ours. Instead, the cut-off value that was established for the presence of CAT was 0.31 (92.1% Se, 66% Sp, AUROC 0.828). It was significantly lower than the cut-off value that we achieved (>1). In addition, they discovered that there was no association between SR and TSH, but that there was a correlation between SR and ATPO results, in accordance with our results [32]. 

The cut-off value for CAT diagnosis was recommended to be >0.98 another study that included 76 adolescents with HT and 46 adolescents without thyroid pathology. This result corresponds to the cut-off value that we obtained (>1; AUROC = 0.850, Se = 82.2%, Sp = 87.5%). The mean values for children with CAT were 1.2 ± 0.2, which was considerably higher than the mean SR values for control participants, which were 0.77 ± 0.18 (*p* 0.01). These results are comparable to our own findings [33].

In yet more research, the effectiveness of SE in the examination of children with thyroiditis was established by the inclusion of 52 children who had been diagnosed with CAT and 22 healthy children. When compared with the CAT group, the control group had considerably lower values for the mean SR (0.68 ± 0.2 vs. 1.19 ± 0.25; *p* < 0.0001). It was observed that a mean value above 0.9 was predictive for CAT, with a sensitivity of 84.62%, a specificity of 95.45%, a positive predictive value of 97.8%, a negative predictive value of 72.4%, and an area under a receiver operating characteristic (AUROC) of 0.9 [34].

Although there is a greater number of studies regarding SWE than there are about SE, the number of SWE studies on children is still significantly lower than the number of adult population studies.

The elasticity scores of 107 healthy children were analyzed in a study, and the researchers determined that the median value of 6.38 ± 1.97 kPa should be considered the standard norm for non-pathological thyroid. Based on our research, we discovered that healthy children have much higher values, with a median EI of 10.1 (9.3–11.2) kPa [35].

A cut-off value of 12 kPa for the median SWE value was determined in this study for the purpose of predicting CAT. This cut-off value had a sensitivity of 88.9% and a specificity of 93.7% according to the statistical analysis that was performed. Research that involved 59 children who had been diagnosed with CAT as well as 26 healthy volunteers led to the same conclusions. The cut-off value for elasticity readings was determined to be 12.3 kPa (Se 86.4, Sp 96.3%). This value had the best diagnostic accuracy possible [36].

Similar cut-off values were obtained in the study that was carried out by Hazel et al. (cut-off > 12.8 kPa with a sensitivity of 87.84%, specificity of 90%, positive predictive value of 97.01%, and 66.7% negative predictive value) [37], but also by Cepeha et al. (cut-off > 12.2 kPa, AUROC −0.88, Se −82%, Sp −88%, PPV −87%, NPV −83%) [38].

Similar findings were discovered by a team of Egyptian researchers who carried out their investigation on 64 young children as part of their study. Patients diagnosed with CAT had elasticity values that were considerably greater (35.6 kPa, with an interquartile range of 8.43–103.7 kPa) than the control group (9.35 kPa, with an interquartile range of 5.73–13.21 kPa). The threshold value for elasticity was determined to be 12.317 kPa, and the sensitivity and specificity of the test were 96.9% and 100%, respectively [39]. 

Koca et al. compared 46 people who were newly diagnosed with CAT to 46 people who were healthy. When compared to healthy controls, individuals with euthyroid CAT had mean SWE levels of 12.5 ± 5.1 kilopascals (kPa), while healthy controls had mean SWE values of 8.2 ± 2.82 kPa (*p* < 0.001). They found a SWE cut-off value of 9.68 kPa, which had a sensitivity of 68%, a specificity of 72%, a 70% positive predictive value, and a 69% negative predictive value. Additionally, this value had a positive predictive value of 70% and a negative predictive value of 69%. Although the mean results obtained are similar to ours, the cut-off value is slightly different than what we obtained, with lower indicators as well [40]. 

If the present investigations are approximately in agreement with one another in terms of the cut-off values, the results are not in accordance with one another in terms of the correlations between the level of elasticity established by elastography and the different variables.

We observed a strong correlation between EI and ATPO levels, and a moderate correlation between EI and ATG levels. Various results were found in the literature. While some studies have found a mild correlation between EI and ATG and a moderate correlation with ATPO [37], others have found no correlation between EI and antibody levels [41,42]. In addition, Kamel et al. found correlations between elasticity and antibody levels to be statistically insignificant (*p* > 0.05) [39]. Kandemirli et al. found a moderate correlation between elasticity and ATPO, but a non-significant correlation between EI and ATG (*p* > 0.05) [36]. Regarding correlations between EI and thyroid functional tests (TSH, FT4), we found no correlations. The majority of data found in specialized literature was consistent with our findings. In several other studies, no correlation with TSH was discovered [36,41,43]. However, there is a study that found a very weak correlation between the level of thyroid elasticity and the TSH level [37].

Regarding strain elastography, we noted a moderate correlation between SR and ATPO and a very weak correlation between SR and ATG. SR was found correlated with ATPO in another study conducted 76 patients diagnosed with CAT and 46 healthy controls, but no correlation was found between SR and ATG [33]. Similar results were obtained by Ozturk et al. [32] wherein SR was correlated with ATPO also in some studies conducted on adult patients with CAT [44]. We found no correlation between SR and TSH or FT4, in accordance with literature findings [32]. There was also no correlation found between SR and age (r = 0.14) or gender (r = 0.19). Similar results were obtained by Yurttutan et al. when analyzing a group of healthy children [31].

Another aspect of our study was the comparison of thyroid elasticity in children diagnosed with CAT who needed treatment and euthyroid patients. We found no statistically significant differences. There are studies on the adult population that have identified differences between patients who receive treatment and those who do not [45]. Thus, elastography may be able to identify a more advanced stage of the disease that is associated with hypofunction, which could be an important discovery. There are currently no further US or US-based characteristics that can discriminate between patients with euthyroid-state CAT and CAT patients with hypothyroidism. Therefore, this line of investigation should be pursued further.

## 5. Conclusions

According to the results of our study comparing strain elastography and shear-wave elastography in children with CAT, both techniques appear helpful for assessing the elasticity of thyroid tissue. SWE proved to be more sensitive and more specific compared to SE, but there are no statistically significant differences between the diagnostic accuracy and dependability of these two methods, so both strain elastography and shear-wave elastography could be useful methods for clinicians evaluating the thyroid health of children. Given the lack of data for this population subgroup, the results obtained are significant, encouraging, and pave the way for future, more extensive research. This research supports elastography for the evaluation of children suspected of having autoimmune thyroiditis, when available.

## Figures and Tables

**Figure 1 biomedicines-11-01970-f001:**
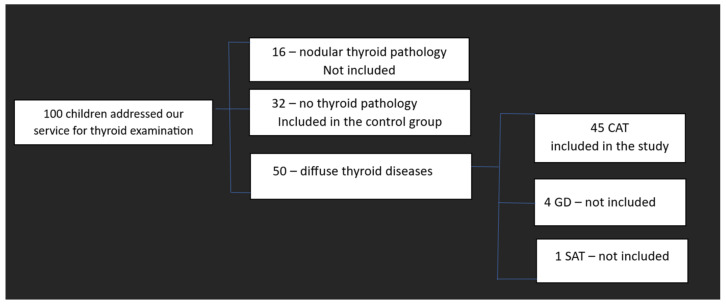
The recruitment process of the subjects. CAT—Chronic autoimmune thyroiditis; GD-Graves’ disease; SAT—subacute thyroiditis.

**Figure 2 biomedicines-11-01970-f002:**
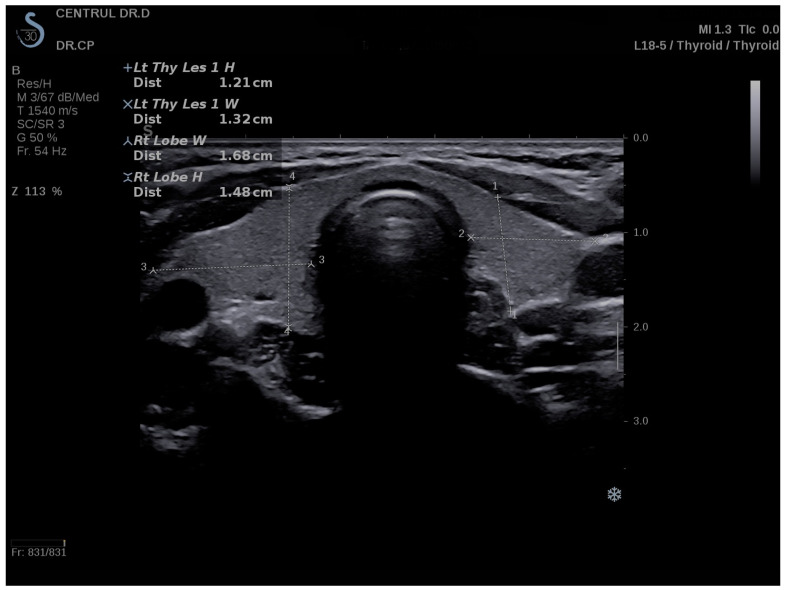
Conventional ultrasound of the thyroid, volume measurements—transverse plane.

**Figure 3 biomedicines-11-01970-f003:**
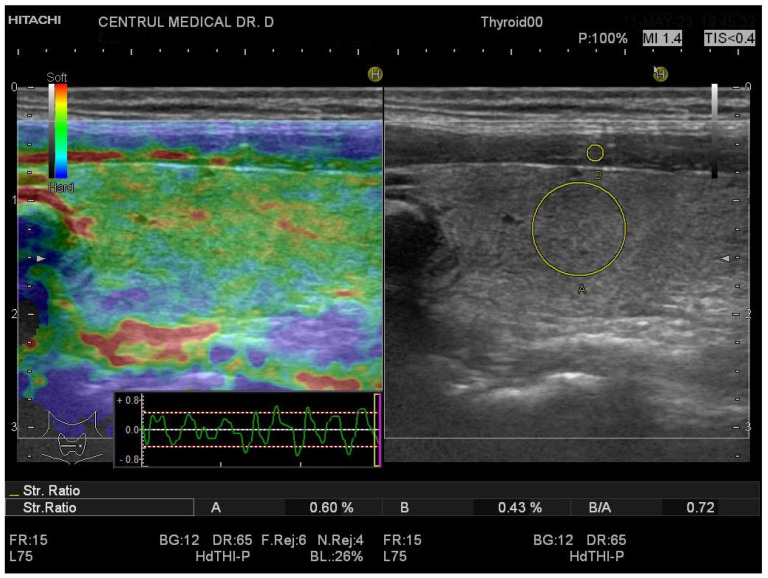
Strain elastography of the thyroid- longitudinal plane, girl, 16 years old, no thyroid pathology SR = 0.72.

**Figure 4 biomedicines-11-01970-f004:**
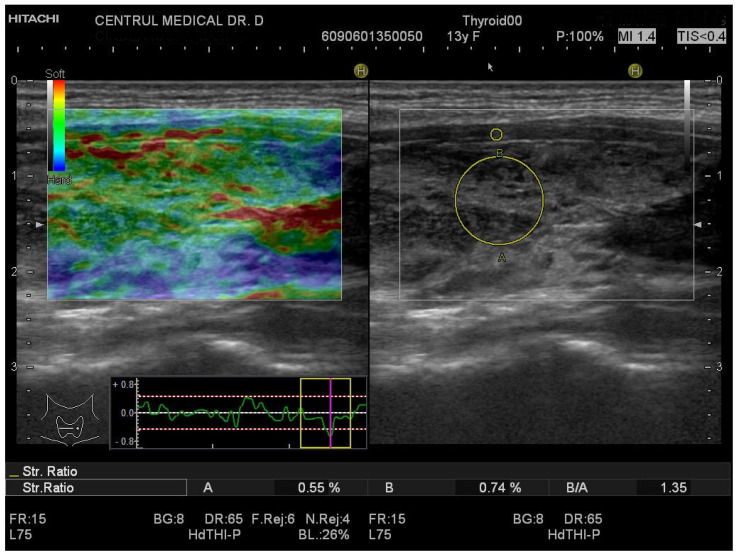
Strain elastography of the thyroid- longitudinal plane, girl, 13 years old, CAT. SR = 1.35.

**Figure 5 biomedicines-11-01970-f005:**
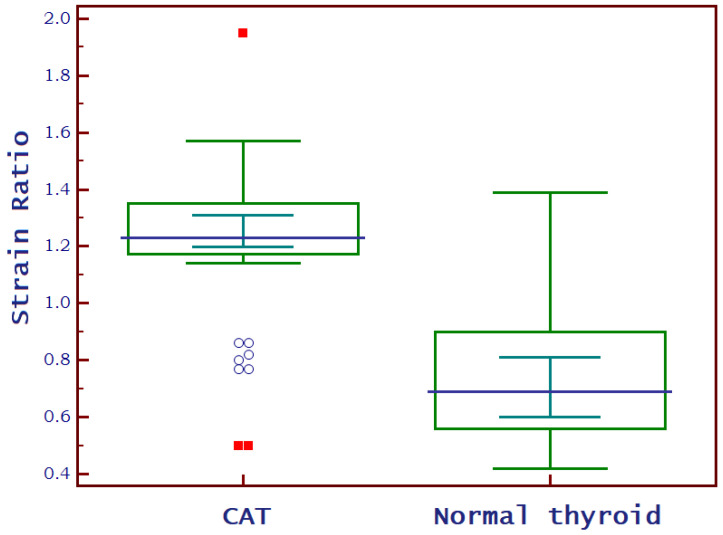
Box-and-whisker distribution plots comparing SR of children diagnosed with CAT to those with normal thyroid.

**Figure 6 biomedicines-11-01970-f006:**
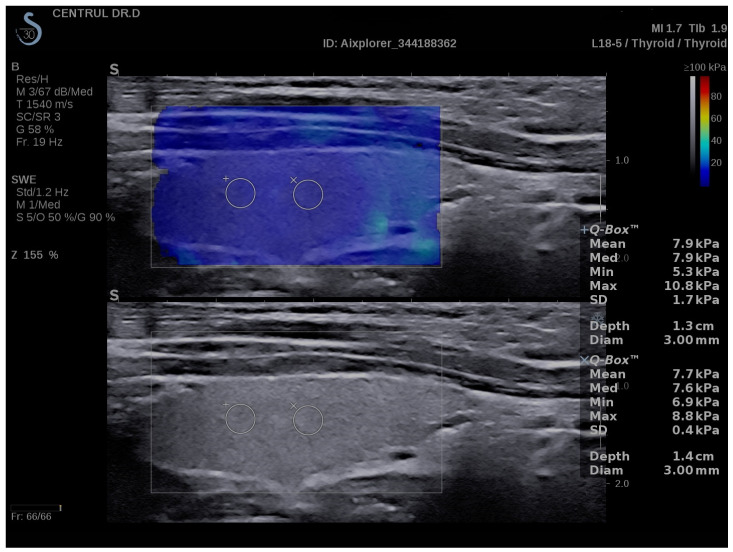
SWE measurements in a normal thyroid- longitudinal plane, girl, 16 years old.

**Figure 7 biomedicines-11-01970-f007:**
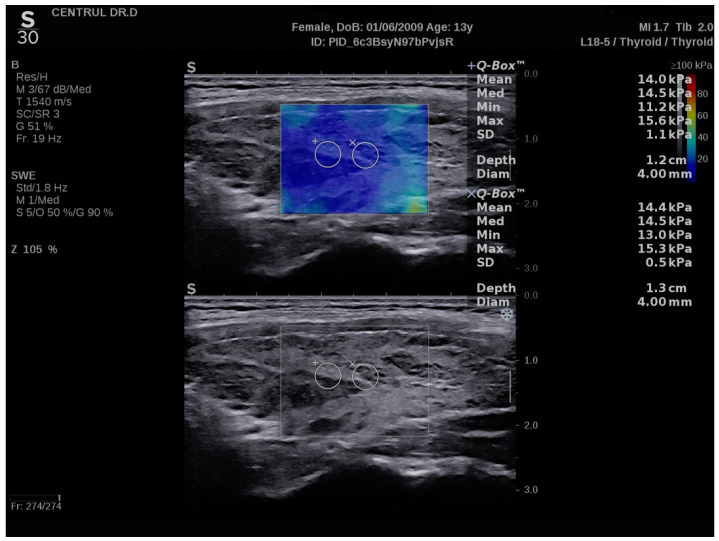
SWE measurements of the thyroid-longitudinal plane, 13 years old girl, CAT.

**Figure 8 biomedicines-11-01970-f008:**
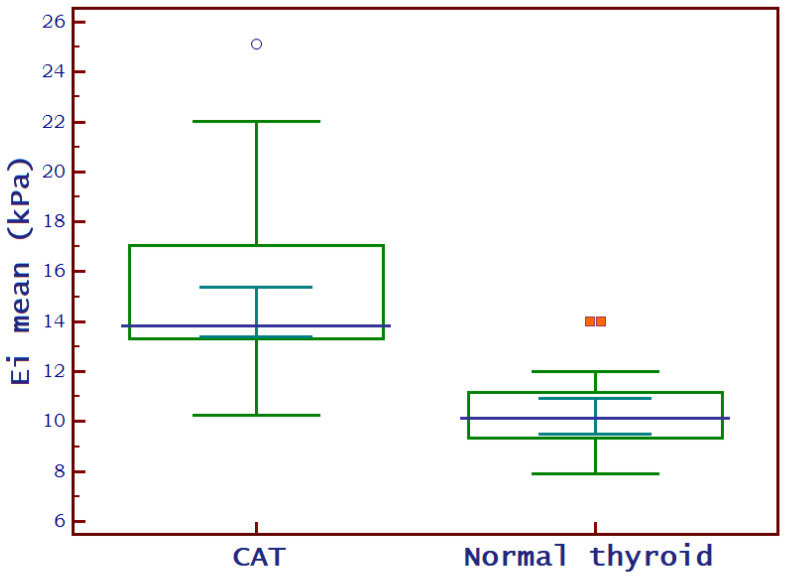
Box-and-whisker distribution plots comparing EI of children diagnosed with CAT to those with normal thyroid. EI = mean elasticity index measured in kilo-Pascals; CAT = chronic autoimmune thyroiditis.

**Figure 9 biomedicines-11-01970-f009:**
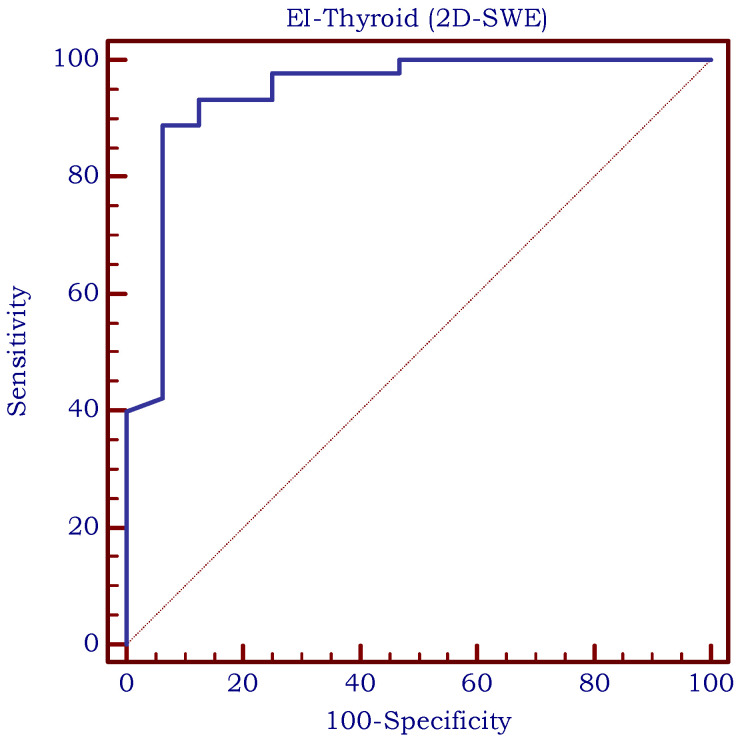
The area under the receiver operating characteristic curve (ROC-curve) for SWE measurements. EI = elasticity index measured using 2D-shear-wave elastography.

**Figure 10 biomedicines-11-01970-f010:**
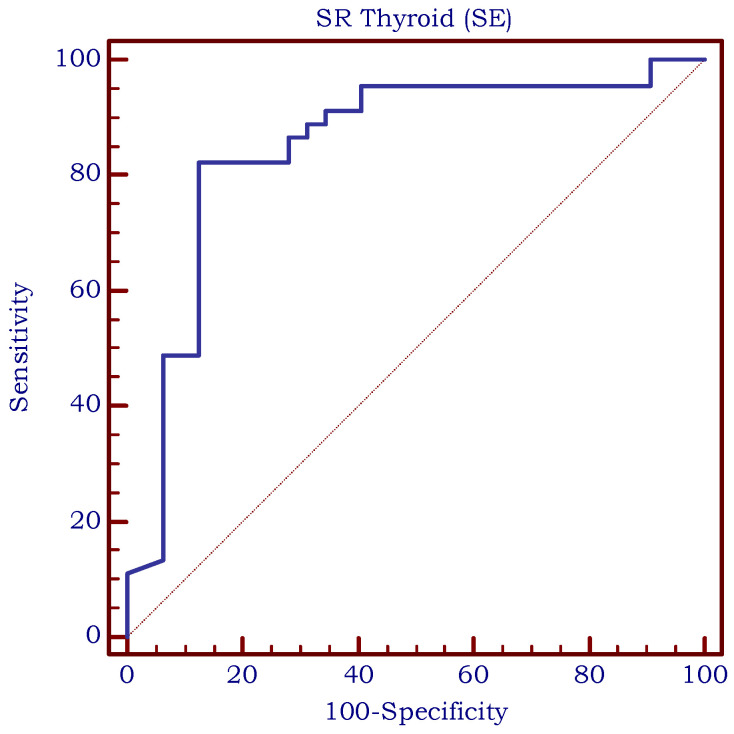
The area under the receiver operating characteristic curve (ROC-curve) for SR measurements; SR = strain ratio measured using strain elastography.

**Table 1 biomedicines-11-01970-t001:** Baseline characteristics.

	Entire Group	CAT	Control	*p*
Number of patients	77	45	32	-
Age	12.9 ± 2.9	12.9 ± 3.1	12.9 ± 2.7	1
BMI	19.6 (18–23.2)	20.1 (18.3–23.3)	18.6 (17.3–20.2)	0.34
TSH	3.27 (2.2–4)	3.4 (1.9–4)	3.2 (2.4–3.9)	0.752
FT4	13.7 (12.6–15.1)	14.5 (12.8–15.1)	13.6 (12.4–14.7)	0.131
ATPO	75 (11.5–542)	476 (203–755)	11 (9–29.5)	<0.0001
ATG	36 (10–170.2)	120 (64.5–288)	14.6 (8.8–27.9)	<0.0001
TV	12.6 ± 6	14.9 ± 6.5	9.3 ± 3	<0.0001
Levothyroxine	-	0 (0–37.5)	-	-

CAT—chronic autoimmune thyroiditis; BMI—body mass index; TSH—thyroid-stimulating hormone; FT4—free thyroxine; ATPO—antithyroid peroxidase antibodies; ATG—antithyroglobulin antibodies; TV—thyroid volume.

**Table 2 biomedicines-11-01970-t002:** Elastography measurements in the two subgroups.

		CAT	Control	*p*
SWE	Mean EI-THY (kPa)	13.8 (13.3–17)	10.1 (9.3–11.2)	<0.0001
	Mean EI-LL (kPa)	13.8 (13.1–17)	10.4 (9.1–11.2)	<0.0001
	Mean EI-RL (kPa)	13.9 (13.3–16.6)	10.2 (9.4–11.1)	<0.0001
SE	SR-THY	1.2 (1.2–1.3)	0.7 (0.6–0.9)	<0.0001
	SR-LL	1.3 (1–1.5)	0.7 (0.5–0.8)	<0.0001
	SR-RL	1.2 (1–1.4)	0.7 (0.6–1)	<0.0001

SWE—shear-wave elastography; SE—strain elastography; EI—elastic index; THY—thyroid; LL—left lobe; RL—right lobe; SR—strain ratio.

**Table 3 biomedicines-11-01970-t003:** Differences in thyroid elasticity between the CAT subgroup with and without thyroid replacement therapy.

	LT4 Treatment	Without Treatment	*p*
EI mean (kPa)	13.4 (13.3–15.5)	14.5 (13.5–17.4)	0.118
SR	1.2 (0.9–1.3)	1.2 (1.2–1.4)	0.290

EI—elastic index; SR = strain ratio; LT4—levothyroxine.

**Table 4 biomedicines-11-01970-t004:** Correlations between different parameters.

		ATG	ATPO	CAT	EI-Mean	LT4	FT4	Gender	BMI	SR-Mean	TSH	Age
ATPO	Correlation Coefficient Significance Level *p* n	0.529 <0.0001 77										
CAT	Correlation Coefficient Significance Level *p* n	0.578 <0.0001 77	0.834 <0.0001 77									
EI-mean	Correlation Coefficient Significance Level *p* n	0.390 0.0004 77	0.665 <0.0001 77	0.757 <0.0001 77								
LT4	Correlation Coefficient Significance Level *p* n	−0.040 0.8724 19	−0.374 0.1151 19	0.000 *p*- 19	−0.211 0.3861 19							
FT4	Correlation Coefficient Significance Level *p* n	0.430 0.0001 77	0.198 0.0841 77	0.173 0.1320 77	0.160 0.1636 77	0.237 0.3283 19						
Gender	Correlation Coefficient Significance Level *p* n	−0.073 0.5254 77	0.107 0.3558 77	0.088 0.4481 77	0.104 0.3660 77	0.176 0.4718 19	−0.247 0.0303 77					
BMI	Correlation Coefficient Significance Level *p* n	0.271 0.0177 77	0.154 0.1847 77	0.244 0.0336 77	0.269 0.0186 77	−0.535 0.0182 19	−0.147 0.2066 77	−0.113 0.3309 77				
SR-mean	Correlation Coefficient Significance Level *p* n	0.264 0.0203 77	0.465 <0.0001 77	0.598 <0.0001 77	0.725 <0.0001 77	−0.128 0.6028 19	0.170 0.1398 77	−0.194 0.0901 77	0.309 0.0065 77			
TSH	Correlation Coefficient Significance Level *p* n	0.104 0.3702 77	−0.070 0.5448 77	−0.036 0.7547 77	−0.093 0.4191 77	0.011 0.9651 19	0.099 0.3928 77	−0.184 0.1097 77	0.225 0.0512 77	−0.022 0.8486 77		
Age	Correlation Coefficient Significance Level *p* n	0.306 0.0067 77	0.051 0.6572 77	0.029 0.8046 77	0.145 0.2092 77	0.134 0.5834 19	0.043 0.7122 77	−0.063 0.5856 77	0.394 0.0004 77	0.145 0.2083 77	0.089 0.4404 77	
TV	Correlation Coefficient Significance Level *p* n	0.430 0.0001 77	0.432 0.0001 77	0.438 0.0001 77	0.434 0.0001 77	−0.275 0.2548 19	0.095 0.4124 77	0.038 0.7416 77	0.407 0.0003 77	0.300 0.0080 77	−0.113 0.3277 77	0.385 0.0006 77

CAT—chronic autoimmune thyroiditis; BMI—body mass index; TSH—thyroid-stimulating hormone; FT4—free thyroxine; ATPO—antithyroid peroxidase antibodies; ATG—antithyroglobulin antibodies; TV—thyroid volume; EI—elastic index; SR—strain ratio; LT4—levothyroxine; n—number of subjects.

## Data Availability

The data presented in this study are available on request from the corresponding author. The data are not publicly available due to patient privacy IRB requirement.

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
