# Peer review of "A Comparative Analysis of Strain and 2D Shear Wave Elastography in the Diagnosis of Autoimmune Thyroiditis in Pediatric Patients"

_biomedicines, 2023, doi:10.3390/biomedicines11071970_

Round 1
Reviewer 1 Report
This paper deals with the imaging of the thyroid gland using elastogaphy. This is now an increasingly used method for the examination of parenchymal organs in particular. The number of patients is sufficient. The methodology is appropriate. The literature is appropriately selected.
[91[ please include the abbreviation CAT in brackets, as it is used later,
[91] please change ATG to aTG
[117] please change ATPO to aTPO
[138] whether the test was carried out by the same person and what his/her qualifications were
[260] Fig. 8 is poorly readable. How about spreading the diagrams over two figures?
Author Response
Dear Reviewer 1,
Your meticulous attention to detail and constructive comments have not only helped me address certain areas that needed further development but have also broadened my perspective on the subject matter. Your expertise and suggestions have undoubtedly enhanced the overall clarity and coherence of my article.
- [91[ please include the abbreviation CAT in brackets, as it is used later,
Thank you for your observation, the abbreviation was added.
- [91] please change ATG to aTG
[117] please change ATPO to aTPO
Thank you for your attention, we made the3 suggested changes.
- [138] whether the test was carried out by the same person and what his/her qualifications were
Thank you for your remark. We added this information in the manuscript: “All measurements were performed by the same practitioner with an experience of at least 3 years in thyroid imaging.”
- [260] Fig. 8 is poorly readable. How about spreading the diagrams over two figures?
We split Fig 8 in 2 different figures, now fig 9 and 10.
Reviewer 2 Report
The authors describe a study comparing two elastography methods for the diagnosis of autoimmune thyroiditis in children aged 6 to 18 years. Despite the diagnostic interest of these techniques, there are several very relevant points to consider:
1. Why SWE or SE should be justified, what about the other techniques, why these would be the most suitable for the study? This should be justified in more detail in the introduction.
2. I would recommend quantifying prevalence in rate or with incidences in specific time periods and precise populations.
3. The study design and sample are not detailed. The number of cases must meet a number of requirements.
4. Why is a non-normal test assumed for quantitative measures in the design, why the division into groups, why is a shapiro wilk test not used? Where is the fisher exact test used?
5. Table 1 shows p= 1 for age, 1 or 0 should not appear.
6. It is recommended not to include height and weight if there is BMI, it could be redundant.
7. How is the diameter of the ROIs justified? The ultrasound settings are also not detailed in terms of the exact frequency used.
8. In the study, there are no differences before or after treatment, why not do an ANOVA or one of its variants for pre-post?
9. What is the correlation between CAT and LT4? P-? Furthermore, correlating everything with everything is sought without physical justification or medical interpretation of possible associations.
10. The conclusion does not seem so justified knowing that SWE is much more sensitive. One would have to test several configurations to be sure that there is no difference.
I did not find it difficult to read the manuscript, but there are sometimes colloquial and ambiguous expressions, and I would recommend a general review.
Author Response
Dear Reviewer 2,
I truly appreciate the expertise and attention to detail you brought to the review process. Your constructive feedback and suggestions have proven instrumental in refining and strengthening the content of my article. Your keen observations and recommendations have undoubtedly elevated the overall clarity and impact of the research, and I am immensely grateful for your guidance.
These are the answers to your comments and suggestions:
- Why SWE or SE should be justified, what about the other techniques, why these would be the most suitable for the study? This should be justified in more detail in the introduction.
SWE and SE are the 2 main categories of elastography used for thyroid assessment. Transient (1D) elastography, also known as Fibroscan is not suitable for thyroid examination. Regarding SWE, studies have shown that 2D-SWE is more accurate than pSWE. We chose to perform thyroid elastography using 2 high-end variants of these two types of elastography provided by Hitachi and Supersonic.
Sigrist RMS, Liau J, Kaffas AE, Chammas MC, Willmann JK. Ultrasound Elastography: Review of Techniques and Clinical Applications. Theranostics 2017; 7(5):1303-1329. doi:10.7150/thno.18650. https://www.thno.org/v07p1303.htm
- I would recommend quantifying prevalence in rate or with incidences in specific time periods and precise populations.
Thank you for your comment. Incidence and prevalence vary depending on location, but we added some specific examples.
- The study design and sample are not detailed. The number of cases must meet a number of requirements.
We added made some adjustments, thank you for your comment.
- Why is a non-normal test assumed for quantitative measures in the design, why the division into groups, why is a shapiro wilk test not used? Where is the fisher exact test used?
Having a small sample of children, we chose to use D’agostino Pearson test to determine if the data follows a normal distribution. At your suggestion, we redid this part of the analysis and applied the Shapiro Wilks test, and the results remained the same. Hence, for non-parametrical variables, we used Mann-Whitney tests and Spearman’s correlations.
- Table 1 shows p= 1 for age, 1 or 0 should not appear.
The p values of 1 is explained by the fact that the two groups are age-matched.
- It is recommended not to include height and weight if there is BMI, it could be redundant.
Thank you very much for your remark. Height and weight were eliminated from the table.
- How is the diameter of the ROIs justified? The ultrasound settings are also not detailed in terms of the exact frequency used.
At this moment there is no reference of guidelines in selecting ROI so we chose to place the ROI in an area in the center of thyroid lobe avoiding measuring marginal areas that could be artefacted or areas that could be too profound. In SWE the ROI measured 3-4 mm, but in STRAIN, the ROI can’t be measured.
The frequency used was 15 MHz, thank you for your observation, we added this in the manuscript.
- In the study, there are no differences before or after treatment, why not do an ANOVA or one of its variants for pre-post?
Our study is not a longitudinal study, we don’t have data before after treatment.
We included different patients with CAT, some of them not receiving treatment because they are euthyroid patients, but other did receive treatment.
Anyway, thank for suggestion, this could be a great idea for research in the future.
- What is the correlation between CAT and LT4? P-? Furthermore, correlating everything with everything is sought without physical justification or medical interpretation of possible associations.
The CAT group is divided into two other subgroups: children receiving LT4 treatment and euthyroid children who didn’t need treatment. It is not relevant to correlate the presence of CAT with the administration of LT4 because no patient from the control group is receiving treatment, only some of the patients with CAT. Indeed, it’s redundant, but we didn’t explain in text, so we added this information. Thank you for your advice.
- The conclusion does not seem so justified knowing that SWE is much more sensitive. One would have to test several configurations to be sure that there is no difference.
Thank you for your comment. Both methods were found to be effective in assessing children with CAT, but we modified the conclusion in order to be more specific about the differences between the two elastography methods.
Reviewer 3 Report
I think this is a solid piece of work and well done. The authors have addressed a narrow question, comparing SE to SWE, and have done it well. It is helpful to know that the two techniques are equivalent and having a sense of the appropriate cut-off points between abnormal and normal. I think it's adequate to publish "as is" with some minor English language copyediting.
Occasional minor mistakes; easily corrected.
Author Response
Dear Reviewer 3,
I wanted to extend my heartfelt appreciation for taking the time to review my article. Your thorough examination and positive evaluation of my work have been truly encouraging. Your assessment of the article and your acknowledgment of its quality as it stands have boosted my confidence in the research and writing I have put forth. Your feedback reinforces my belief in the importance of this topic and the merit of the research. Your review has served as a valuable validation of my efforts, and I am honored by your assessment.